# Minimally Invasive versus Full Sternotomy SAVR in the Era of TAVR: An Institutional Review

**DOI:** 10.3390/jcm11030547

**Published:** 2022-01-22

**Authors:** Tyler W. Wilson, Joshua J. Horns, Vikas Sharma, Matthew L. Goodwin, Hiroshi Kagawa, Sara J. Pereira, Stephen H. McKellar, Craig H. Selzman, Jason P. Glotzbach

**Affiliations:** 1School of Medicine, University of Utah, Salt Lake City, UT 84132, USA; tyler.w.wilson@hsc.utah.edu; 2Department of Surgery, University of Utah, Salt Lake City, UT 84132, USA; joshua.horns@hsc.utah.edu; 3Division of Cardiothoracic Surgery, University of Utah, Salt Lake City, UT 84132, USA; vikas.sharma@hsc.utah.edu (V.S.); matt.goodwin@hsc.utah.edu (M.L.G.); hiroshi.kagawa@hsc.utah.edu (H.K.); sara.pereira@hsc.utah.edu (S.J.P.); craig.selzman@hsc.utah.edu (C.H.S.); 4Cardiovascular and Thoracic Surgery, Intermountain Medical Center Heart Institute, Salt Lake City, UT 84107, USA; stephen.mckellar@imail.org

**Keywords:** minimally invasive, aortic valve replacement, hemi-sternotomy, median sternotomy

## Abstract

In the era of advancing transcatheter aortic valve replacement (TAVR) technology, traditional open surgery remains a valuable intervention for patients who are not TAVR candidates. We sought to compare perioperative variables and postoperative outcomes of minimally invasive and full sternotomy surgical aortic valve replacement (SAVR) at a single institution. A retrospective analysis of 113 patients who underwent isolated SAVR via full sternotomy or upper hemi-sternotomy between January 2015 and December 2019 at the University of Utah Hospital was performed. Preoperative comorbidities and demographic information were not different among groups, with the exception of diabetes, which was significantly more common in the full sternotomy group (*p* = 0.01). Median procedure length was numerically shorter in the minimally invasive group but was not significant following the Bonferroni correction (*p* = 0.047). Other perioperative variables were not significantly different. The two groups showed no difference in the incidence of postoperative adverse events (*p* = 0.879). As such, minimally invasive SAVR via hemi-sternotomy remains a safe and effective alternative to full sternotomy for patients who meet the criteria for aortic valve replacement.

## 1. Introduction

With increased advancements in transcatheter aortic valve replacement (TAVR) technology, its implementation is continually expanding as a treatment for aortic valve pathology. Although the PARTNER and SURTAVI trials demonstrated good results that support the increased utilization of TAVR, certain patient populations, including those with bicuspid aortic valves and mixed aortic valve disease (insufficiency with stenosis), were excluded from the trial data [1,2]. Although TAVR is now widely used as first-line therapy for aortic valve stenosis, not all patients requiring aortic valve intervention are candidates for TAVR. Traditional open surgical aortic valve replacement (SAVR) remains an important therapeutic modality for patients who have either a history of bicuspid aortic valve, mixed pathology, or large annuli and have low-to-intermediate surgical risk [3,4]. Full median sternotomy for SAVR has been the routine approach since the 1960s, demonstrating consistently low morbidity at experienced centers [5]. Despite these consistent outcomes, minimally invasive techniques for SAVR have gained acceptance in the cardiac surgical community as a safe and comparable alternative to conventional methods since their introduction in the late 1990s [6]. As treatment for aortic valve disease moves toward less invasive approaches with TAVR, it is important to continue to assess the outcomes of less invasive surgical approaches, which remain relevant to patients for whom TAVR is not an option.

Studies comparing minimally invasive surgical techniques to full median sternotomy have demonstrated shortened intensive care unit length of stay, shortened total hospital length of stay, and a lower incidence of postoperative complications [7,8,9,10]. The minimally invasive technique offers the additional potential benefits of decreased postoperative bleeding, reduced transfusion requirements, reduced sternal trauma, and an accelerated rehabilitation and return to normal activity [7,8,11]. Based on these and other claims of higher patient satisfaction, many believe that minimally invasive approaches to SAVR offer a safe and effective alternative to conventional full sternotomy [9].

In contrast, other studies have shown that minimally invasive approaches provide poor exposure, leading to decreased visibility, increased cardiopulmonary bypass and aortic cross-clamp time, and longer operations [9,12]. Several authors have suggested that the advantages of minimally invasive surgery are more cosmetic than clinical [9,13,14,15]. The objective of this study is to compare perioperative variables and postoperative outcomes after minimally invasive SAVR via a hemi-sternotomy with those via a full sternotomy at a single institution during the era of TAVR.

## 2. Materials and Methods

### 2.1. Patient Selection

This study is a retrospective review of 113 patients who underwent isolated SAVR for a diagnosis of aortic stenosis or insufficiency at the University of Utah Hospital between January 2015 and December 2019 using one of two techniques: (1) minimally invasive upper hemi-sternotomy (MI) or (2) full sternotomy (FS). To minimize selection bias and interrogate a clinically homogenous cohort, we excluded all patients with a diagnosis of infective endocarditis and those who underwent simultaneous operations, as these more complex and higher-risk patients were much more likely to receive an FS approach.

Nine surgeons performed the operations included in this study. The choice of surgical approach was determined by each surgeon. MI is considered in all patients who require isolated cardiac valve surgical procedures. If the patient meets the criteria for a minimally invasive approach, MI is the preferred technique at our institution. Anterior thoracotomy approaches for SAVR are not performed at our institution.

### 2.2. Data Collection

All preoperative information, in-hospital outcomes, and follow-up outcomes were collected from the electronic medical records of the University of Utah. Institutional Review Board review was waived due to the retrospective nature of the study.

Demographics, preoperative comorbidities, and primary valvular pathology were aggregated, along with cardiopulmonary bypass, aortic cross-clamp, and total operative times; operative morbidity and mortality; length of hospital stays; echocardiogram findings; and postoperative events. Patient information was reviewed up to 30 days postdischarge except for echocardiogram findings, which were recorded at extended postoperative follow-up intervals. Length of stay in this study accounts for the time from the patient’s primary surgical intervention to their discharge from the hospital. Some patients were admitted to the hospital before surgery, but this time was excluded from our data analysis to maintain uniformity among groups. Length of stay was separated into length of intensive care unit (ICU) stay and total hospital length of stay. Postoperative adverse events were compiled and included stroke, bleeding requiring transfusion, deep sternal wound infection, acute kidney injury, 30-day hospital readmission, and 30-day mortality. Postoperative atrial fibrillation and aortic reintervention were evaluated as separate categories.

### 2.3. Surgical Procedure

At our institution, FS is the most common approach for patients undergoing isolated SAVR. However, as surgical techniques have evolved, we have increasingly adopted MI as an alternative approach. The FS approach was performed according to the standard technique, while the MI approach was performed via an upper hemi-sternotomy in an upside-down T configuration through the fourth intercostal space, as previously described [16]. Cardiopulmonary bypass (CPB) was established through direct ascending aorta or femoral arterial cannulation and direct right atrial or percutaneous femoral venous cannulation. Cardioplegia was administered in a combination of antegrade and retrograde or via direct administration into the coronary ostia. A left ventricular vent was placed through the aortic valve or right superior pulmonary vein. The size and choice of valve, as well as implantation technique, were determined at the discretion of the attending surgeon. After surgery, all patients were transferred to the cardiovascular ICU or surgical ICU and managed according to approved postoperative care pathways.

### 2.4. Statistical Analysis

We considered nine outcomes that were each modeled against surgery type using mixed-effect regression: procedure length, cardiac bypass time, cross-clamp time, aortic gradient, length of stay (both ICU and overall), aortic reintervention, atrial fibrillation, and postoperative events. Procedure length, cardiac bypass time, cross-clamp time, and aortic gradient were modeled using linear regression. LOS was modeled using Poisson regression. Aortic reintervention, atrial fibrillation, and postoperative events were modeled using logistic regression. In all models, surgeon was included as a random effect. We performed a Bonferroni correction to account for multiple testing of each outcome.

## 3. Results

Patient demographics, comorbidities, and preoperative characteristics are described in Table 1.

Of the 113 operations performed during our study period, 60 (53.1%) used the FS approach and 53 (46.9%) used the MI approach (Table 1). Preoperative characteristics for the two groups were similar, except that 20 (33.3%) of the FS patients reported diabetes as a comorbid condition compared to only 6 (11.1%) of the MI patients; the difference was statistically significant (*p* = 0.01). With the exception of overall procedure time, perioperative variables and outcomes were not different between the two groups (Figure 1). Perioperative time intervals were numerically shorter in the MI group (216 vs. 246 min, *p* = 0.048; 111 vs. 120 min, *p* = 0.613; and 76 vs. 82 min, *p* = 0.968 respectively), although the differences in procedure length, cardiopulmonary bypass time, and aortic cross-clamp time between FS and MI cases were not significant (Table 2). ICU and total hospital length of stay were not different between the two groups (Table 2).

Both the MI and traditional FS groups showed a similar incidence of postoperative adverse events (Table 2). There were numerically fewer postoperative events in the MI group, but this difference was not significant (*p* = 0.879). There was a trend toward a reduced incidence of postoperative bleeding requiring transfusion in the MI group, although this result was not statistically significant (*p* = 0.052) (Table 2).

## 4. Discussion

Minimally invasive aortic valve replacement has gained increased acceptance in the adult cardiac surgery community. With improving postoperative results, minimally invasive aortic valve surgery has increasingly been used as an alternative strategy for the treatment of aortic valve disease at our institution when criteria for TAVR have not been met. However, many studies continue to describe longer operative times for minimally invasive techniques, and some continue to question the effectiveness of these surgical techniques [14,17]. In our study, we found that MI SAVR had comparable clinical outcomes to FS SAVR.

Most comparisons of surgical approaches for aortic valve surgery have demonstrated that minimally invasive approaches can provide adequate operative visualization, reduced recovery time, and improved cosmetic results, all while also reducing total hospital cost [7,18,19]. Although some of these studies reported equivalent outcomes between minimally invasive and conventional methods [7,19], many demonstrated improved postoperative outcomes with the use of minimally invasive approaches, including reduced assisted ventilation duration, decreased need for blood product transfusion, shorter hospital stays, and lower mortality [5,6,7,18,20]. While the existence of many retrospective comparisons of the surgical aortic valve techniques is known, it is important to recognize that fewer prospective randomized trials have been performed to evaluate the risk–benefit ratio of minimally invasive aortic surgery [7,10,17,21]. One small trial did report equally safe and reliable outcomes for MI SAVR as compared to FS, but additional larger trials are needed to solidify the claims of improved outcomes with minimally invasive techniques [21].

The results from the current study are consistent with the findings in other studies. We observed a shorter postoperative cardiovascular ICU length of stay and total hospital length of stay following minimally invasive hemi-sternotomy as compared to full sternotomy. Although not statistically significant, this suggests that hemi-sternotomy is comparable to full sternotomy with regard to postoperative hospital course and recovery time.

MI SAVR had no difference in the incidence of postoperative adverse events, including bleeding requiring transfusion, acute renal injury, deep sternal wound infection, and 30-day hospital readmission when compared to the full sternotomy approach. This, in addition to the tendency for shorter ICU and total hospital stay, suggest a move toward improved outcomes and faster recovery with minimally invasive hemi-sternotomy for SAVR at our institution. This finding aligns with the literature that reports reduced ICU time, fewer postoperative complications, and faster recovery with MI approaches [9,10]

Most prior studies have reported higher operative, cardiopulmonary bypass, and aortic cross-clamp times with minimally invasive techniques compared to full sternotomy, while some studies have demonstrated comparable times among groups [5,8,12,13,20]. Other centers have reported reduced cardiopulmonary bypass and aortic cross-clamp times, shorter postoperative length of stay, and decreased complications with minimally invasive AVR [7,10,22,23]. This suggests that increased training and experience with minimally invasive techniques among surgeons improves perioperative times and surgical outcomes.

In our study, the cardiopulmonary bypass time and aortic cross-clamp time were numerically shorter in the MI group, although not at a statistically significant interval. Additionally, the total procedure length was shorter with the hemi-sternotomy approach as compared to the full sternotomy. This likely reflects improvements in team efficiency with patient and operating room preparation, draping, and instrument availability. The decision to perform a minimally invasive operation relies on the comfort level of the attending surgeon. Not all of our surgeons use these techniques. Therefore, this trend toward shorter operative times with the MI approach may be due to the training, experience, and expertise of the surgeons who do perform SAVR via hemi-sternotomy. When surgeon identity was added to the analysis as a random effect, no differences were observed between the two surgical approaches.

Our study analyzed the outcomes of aortic valve replacement performed utilizing a minimally invasive technique (hemi-sternotomy) compared with the standard full sternotomy. The findings of this study have several important limitations. We report a retrospective analysis of a single-center experience with a relatively small sample size (113 patients). In addition, the follow-up time period was limited (30-day follow-up). The inclusion of only one minimally invasive surgical technique can also be considered a limiting factor.

## 5. Conclusions

In the era of TAVR, minimally invasive aortic valve replacement via hemi-sternotomy remains a safe and effective alternative to full median sternotomy, showing no significant difference in mortality or postoperative adverse events. The minimally invasive hemi-sternotomy approach demonstrates reduced operative times and similar lengths of recovery compared to full sternotomy and remains a viable option for patients who meet the criteria for aortic valve replacement.

## Figures and Tables

**Figure 1 jcm-11-00547-f001:**
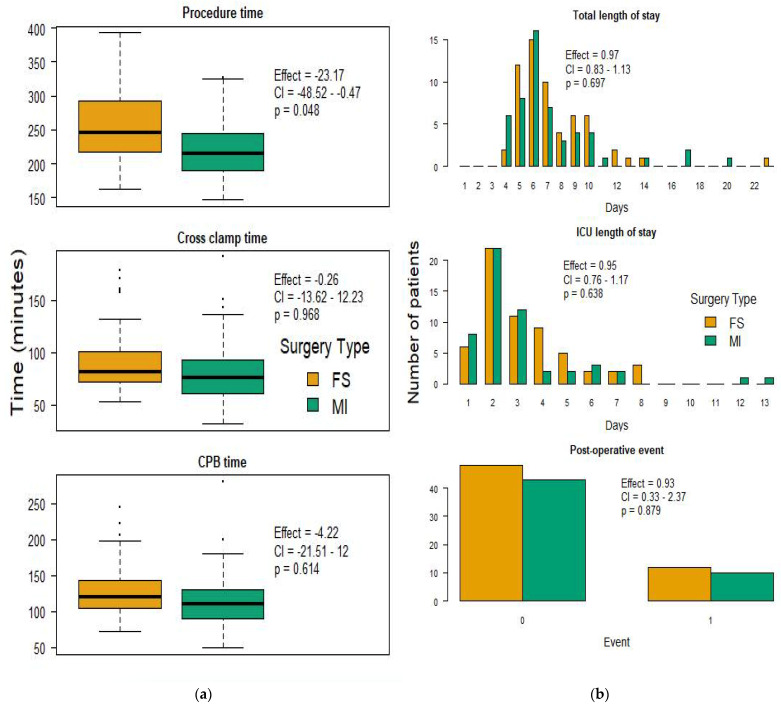
Perioperative time intervals and select postoperative outcomes. (**a**) Mean procedure, cross-clamp, and cardiopulmonary bypass (CPB) times in minutes comparing minimally invasive (MI; green) and full sternotomy (FS; yellow). (**b**) Comparison of patient postoperative intensive care unit (ICU) and total hospital length of stay in days and postoperative events, with 0 indicating no events and 1 representing adverse event occurrence.

**Table 1 jcm-11-00547-t001:** Preoperative characteristics.

	Surgical Approach		
	FS (*n* = 60)	MI (*n* = 53)	*p*-Value
Patients, *n* (%)	60 (53.1)	53 (46.9)	
Age, y (SD)	58.7 (13.19)	60.89 (11.73)	0.353
BMI, (SD)	30.68 (6.89)	29.04 (4.56)	0.136
Gender			1
Male, *n* (%)	39 (65)	34 (64.15)	
Female, *n* (%)	21 (35)	19 (35.85)	
Primary Diagnosis			0.477
AS, *n* (%)	48 (80)	46 (86.79)	
AI, *n* (%)	12 (20)	7 (13.21)	
Preoperative LVEF (SD)	57.7 (10.87)	59.72 (10.4)	0.316
Diabetes, *n* (%)	20 (33.3)	6 (11.32)	0.01
Hypertension, *n* (%)	33 (55)	27 (50.94)	0.81
CAD, *n* (%)	9 (15)	5 (9.43)	0.542
Hyperlipidemia, *n* (%)	26 (43.33)	25 (47.17)	0.826
Renal Failure, *n* (%)	4 (6.67)	3 (5.66)	1
CHF, *n* (%)	4 (6.67)	3 (5.66)	1
Cerebrovascular Disease, *n* (%)	0 (0)	0 (0)	
Previous Stroke, *n* (%)	4 (6.67)	3 (5.66)	1
Current Smoker, *n* (%)	5 (8.33)	6 (11.32)	0.828
Previous Smoker, *n* (%)	22 (36.67)	15 (28.3)	0.456
BAV, *n* (%)	27 (45)	33 (62.26)	0.1

Abbreviations: FS, full sternotomy; MI, minimally invasive; SD, standard deviation; BMI, body mass index; AS, aortic stenosis; AI, aortic insufficiency; LVEF, left ventricular ejection fraction; CAD, coronary artery disease; CHF, congestive heart failure; BAV bicuspid aortic valve.

**Table 2 jcm-11-00547-t002:** Perioperative parameters and postoperative outcomes.

	Surgical Approach		*p*-Value
	FS	MI	
Procedure length (min), median (IQR)	247 (217.25–292)	216 (190–244)	0.048
CPB time (min), median (IQR)	120 (104–142.25)	111 (90–131)	0.613
Cross-clamp time (min), median (IQR)	82 (72–101)	76 (61–93)	0.968
Mean aortic gradient, median (IQR)	11 (9–13.6)	10.7 (9.15–13.73)	0.712
ICU LOS (days), median (IQR)	3 (2–4)	2 (2–3)	0.636
Total hospital LOS (days), median (IQR)	7 (6–9)	6 (5–8)	0.697
Aortic valve reintervention, *n* (%)	1 (1.67)	3 (5.66)	0.280
Atrial fibrillation, *n* (%)	15 (25)	18 (33.96)	0.297
Postoperative event, *n* (%)	12 (20)	10 (18.87)	0.879
Stroke, *n* (%)	0 (0)	2 (3.77)	0.422
Bleeding requiring transfusion, *n* (%)	6 (10)	0 (0)	0.052
Deep sternal wound infection, *n* (%)	1 (1.67)	0 (0)	1
Acute kidney injury, *n* (%)	3 (5)	2 (3.77)	1
30-day readmission, *n* (%)	8 (13.33)	6 (11.32)	0.97
30-day mortality, *n* (%)	0 (0)	1 (1.89)	0.95

Abbreviations: FS, full sternotomy; MI, minimally invasive; IQR, interquartile range; CPB, cardiopulmonary bypass; ICU, intensive care unit; LOS, length of stay.

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
