# Peer review of "Minimally Invasive versus Full Sternotomy SAVR in the Era of TAVR: An Institutional Review"

_jcm, 2022, doi:10.3390/jcm11030547_

Round 1

Reviewer 1 Report

Thank you for the opportunity to review this interesting paper. This paper is comparing outcomes after SAVR via traditional full sternotomy with ones via hemi-sternotomy. The author found similar results between two approaches.

Overall the manuscript is well written but as a reader I wish to make some suggestions that I believe will improve the manuscript further.

Overall:

I don’t think the author can claim that there was a trend favoring the hemi-sternotomy approach based on the p value. Please consider changing the word.

Text:

  1. Line 84-86: I imagined there were not many events after the surgery, but I think the author should report major adverse cardiac events (such as stroke or 30-day mortality in this study) or composite endpoint of 30-day readmission and 30-day mortality separately. In addition, the author should report the details of each events.
  2. Is there any chance to show KCCQ or pain scales after the surgery to understand patients' satisfaction after MI-SAVR comparing to FS-SAVR?
  3. Line 143: Please add a sentence that states the findings from this study and emphasize your message in the beginning of the discussion section for readers' better understanding.
  4. Line 164-165: I don't think the author can claim there is a trend from these results. In terms of ICU length or hospital stay, it may be sutibale to report the results in hours so that you might see differences?
  5. Line 195-196: Is it possible to show the difference in outcomes of MI-SAVR according to surgeons’ expertise? And also, I would be interested in the reasons that expert surgeons deemed patients were not eligible for MI-SAVR and chose FS-SAVR.

Reviewer 2 Report

I enjoyed the opportunity to read "Minimally Invasive SAVR in the Era of TAVR: An Institutional Review".

I would like to congratulate you on such a comprehensive and remarkable study showing comparing outcomes of stented tissue valves in aortic valve replacement interventions. This analysis purports to compare the outcomes of procedures performed by full sternotomy with those done by mini-sternotomy in a historical period in which aortic valve replacement is often performed with trans catheter techniques (TAVI).

Comments and questions are as follows:

- Title dos not accurately describes the manuscript; it is not clear to me the focus of the analysis in comparison with trans-catheter technique;

- the aim of the study is relevant because this phenomenon remains poorly understood and is still debated and analyzed in our literature, but I think that a more accurate analysis must be made of the intra-operative outcomes of aortic valve replacement surgery with the various types of trans-catheter technique in order to be able to compare them to conventional aortic valve replacement operations both in traditional surgery and with minimally invasive technique;

- the sample of patients analyzed is small, about 30 patients per year (113 patients over 5 years). So, could there be an important selection bias? The authors wisely chose to include only low risk patients;

- anamnestic data are mixed with echocardiographic values in Table 1;

- intraoperative values with outcome are mixed in Table 2;

- the number of surgeons who performed the operations is not specified;

- post-operative follow up is only one month;

- Minor remarks: Table 2, for Aortic Valve Reintervention and Atrial Fibrillation, symbol (%) is missing near the values; CBP Parameter and abbreviation in legend (CPB) not alligned (verify also figure 1).

Round 2

Reviewer 2 Report

Revisions are consistent